# A Comprehensive Review on the Techniques for Extraction of Bioactive Compounds from Medicinal Cannabis

**DOI:** 10.3390/molecules27030604

**Published:** 2022-01-18

**Authors:** Hebah Muhsien Sabiah AL Ubeed, Deep Jyoti Bhuyan, Muhammad A. Alsherbiny, Amrita Basu, Quan V. Vuong

**Affiliations:** 1School of Science, College of Sciences, Engineering, Computing Technologies and Health and Medical Sciences, RMIT University, Bundoora, Melbourne, VIC 3083, Australia; 2NICM Health Research Institute, Western Sydney University, Penrith, NSW 2751, Australia; M.Alsherbiny@westernsydney.edu.au; 3Department of Pharmacognosy, Faculty of Pharmacy, Cairo University, Cairo 11562, Egypt; 4Complex Carbohydrate Research Centre, University of Georgia, 315 Riverbend Road, Athens, GA 30602, USA; Amrita.Basu@uga.edu; 5School of Environmental and Life Sciences, College of Engineering, Science, and Environment, The University of Newcastle, 10 Chittaway Road, Ourimbah, NSW 2258, Australia; vanquan.vuong@newcastle.edu.au

**Keywords:** *Cannabis*, medicinal cannabis, cannabinoids, extraction techniques

## Abstract

*Cannabis* is well-known for its numerous therapeutic activities, as demonstrated in pre-clinical and clinical studies primarily due to its bioactive compounds. The *Cannabis* industry is rapidly growing; therefore, product development and extraction methods have become crucial aspects of *Cannabis* research. The evaluation of the current extraction methods implemented in the *Cannabis* industry and scientific literature to produce consistent, reliable, and potent medicinal *Cannabis* extracts is prudent. Furthermore, these processes must be subjected to higher levels of scientific stringency, as *Cannabis* has been increasingly used for various ailments, and the *Cannabis* industry is receiving acceptance in different countries. We comprehensively analysed the current literature and drew a critical summary of the extraction methods implemented thus far to recover bioactive compounds from medicinal *Cannabis*. Moreover, this review outlines the major bioactive compounds in *Cannabis*, discusses critical factors affecting extraction yields, and proposes future considerations for the effective extraction of bioactive compounds from *Cannabis*. Overall, research on medicinal marijuana is limited, with most reports on the industrial hemp variety of *Cannabis* or pure isolates. We also propose the development of sustainable *Cannabis* extraction methods through the implementation of mathematical prediction models in future studies.

## 1. Introduction

There has been an increased interest in medical applications of cannabis over the last decades. *Cannabis* can be classified based on genetics, phenotypic properties, and chemical composition. All these types are rich in bioactive phytochemicals. However, the phytochemical composition varies in different types. For instance, *Cannabis sativa* or industrial hemp has a higher cannabidiol (CBD) level than *Cannabis indica* and *Cannabis ruderalis*, whereas *Cannabis indica* has a higher level of the psychoactive cannabinoid delta-9-tetrahydrocannabinol (Δ^9^-THC), and *Cannabis ruderalis* has a lower level of Δ^9^-THC as compared to *Cannabis sativa* [1]. Most of the cannabinoids in *Cannabis* are derived from cannabigerolic acid (CBGA) via olivetolate geranyl transferase [2]. CBGA is then converted to major cannabinoids, such as Δ^9^-tetrahydrocannabinolic acid (THCA), cannabidiolic acid (CBDA), and cannabichromenic acid (CBCA) by oxide cyclase enzymes, including THCA synthase, CBDA synthase, and CBCA synthase in trichomes [3]. A nonenzymatic reaction caused by drying, heating, or combustion can further produce other active compounds (Figure 1). For instance, tetrahydrocannabinol (THC) can be converted to Δ^8^-THC, cannabinol (CBN), and cannabinolic acid (CBNA). CBD and cannabichromene (CBC) can be converted to cannabicyclolic (CBL), cannabicyclolic acid (CBLA), and cannabigerol (CBG) [4].

*Cannabis* has been traditionally used worldwide as herbal medicine for many years. Recently, commercial products have been developed for convenient uses [5]. In Australia, the two products registered at the Australian of Therapeutic Goods (TGA) are Epidyolex^®^(CBD) and Sativex Oromucosal Spray (nabiximols^®^). Various pharmaceutical products are available in the form of tablets, capsules, and sprays, which can only be obtained via prescription, and unregistered drugs require special approval under the TGA special scheme or Authorised Prescriber Scheme [6]. *Cannabis* is a rich source of phytochemicals with over 125 types of cannabinoids and 400 non-cannabinoids like flavonoids, alkaloids, phenols, and terpenes [7]. These phytochemicals have been linked to various health benefits. For example, Δ^9^-THC and CBD are associated with inflammation and pain reduction [8]. CBD is known to have antipsychotic, anxiolytic, and antiepileptic properties [9]. CBD is also reported to associate with the prevention of Alzheimer’s disease [10], Parkinson’s disease [11], schizophrenia and psychosis [12], and various types of cancer [13]. In addition, other compounds, such as THC, GBG, CBC, and CBN, have also been linked with various health benefits. THC has shown therapeutic activity against arthritic and inflammatory conditions [14], autism [15], Parkinson’s [16], glaucoma [17], and cardiovascular disorders [18,19]. GBG and CBC have strong antibacterial and antifungal effects [20], along with the potential effect of CBN on insomnia and sleep disorders [21]. Despite the current evidence on *Cannabis*, further research is still warranted to explore the interaction of other drugs with medicinal *Cannabis* [22].

The extraction of *Cannabis* phytochemicals is a vital step to separate bioactive molecules from the plant matrix and to enhance the diverse applications of these compounds in the pharmaceutical and food industries. Extraction can be conducted using conventional and advanced techniques. The efficiencies of conventional and advanced methods mainly depend on the critical input parameters. Therefore, understanding the nature of the plant matrix and the chemistry of bioactive compounds is necessary [23]. Conventional and advanced techniques have also been studied to maximise the extraction of phytochemicals from *Cannabis* [24,25]. However, due to the high demand for medicinal *Cannabis* in recent years, a deeper understanding of the *Cannabis* chemical space and how to effectively reduce the costs of extraction and phytochemical isolation is crucial for further applications. This review portrays the knowledge for extraction and isolation of phytochemicals from *Cannabis* by outlining major phytochemicals in *Cannabis*, discussing the factors affecting the extraction efficiencies of phytochemicals, reviewing the current extraction techniques, which have been applied for *Cannabis*-derived phytochemicals, and proposing considerations for future studies.

## 2. Major *Cannabis* Phytochemicals and Their Therapeutic Effects

*Cannabis* exists in a variety of forms; of those, hemp and marijuana are the most well-known types. A hemp variety of *Cannabis* may grow taller and faster, but it does not generate as much THC, the psychoactive ingredient of marijuana. Marijuana strains are rich in THC and CBD and are used for medical and recreational purposes [26]. Different strains of *Cannabis* are available, either pure *Cannabis* genus, including *C. sativa*, *C. indica*, and C. *ruderalis*, or hybrid varieties. Chemotaxonomically, *Cannabis* is classified based on the phytocannabinoids (PCs) content, including THC and CBD in particular [27]. The most commonly grown *Cannabis* species are *sativa* and *indica*, where *ruderalis* is short and produces trace THC amount and is thereby not commonly grown for medicinal or industrial purposes [28]. *C sativa* is an energising strain with common psychoactivity, so it is known for being a ‘head high’, whereas *indica* is more commonly used for relaxation as a sedative and is regarded as ‘body high or couch lock’ [29,30]. 

Over 320 phytochemical compounds have been identified in *Cannabis*; of those, 125 cannabinoids and 198 non-cannabinoids have been found in *Cannabis* (Figure 2) [7]. PCs are produced in the glandular hairs of the *Cannabis* plant (Figures S1–S3) [7]. THC and CBD are the most common psychoactive and non-psychoactive cannabinoids, respectively, which are derived from cannabigerol acid (CBGA) [2] and from the decarboxylation of their respective acids, such as THCA, CBDA, and CBCA (Figure 1). Intriguingly, recent advances in analytical platforms enable the subnanomolar detection and quantitation of cannabinoids in different biological matrices [31]. THCs are the primary psychoactive lipids with an incredibly low water solubility (0.003 mgmL^−1^) [32]. Therefore, various solubilizers or emulsifiers and advanced nanosized drug delivery systems have been applied to increase their bioavailability [33,34,35,36,37,38]. In addition, some formulations are employed to minimise the pharmacokinetic and pharmacodynamic variabilities of cannabinoids [39]. However, different behavioural effects in rats were reported for different pharmaceutical preparations of ∆^9^-THC [40]. 

CBG, CBD, CBND, cannabielsoin (CBE), cannabicyclol (CBL), cannabinol (CBN), cannabitriol (CBT), and cannabichromene (CBC)-like cannabinoids have also been isolated and fully characterised from *Cannabis* extracts. CBG can interact with 5-hyroxytryptamine (5-HT1A) and α-2 adrenoreceptors to have therapeutic potential in Parkinson’s, Huntington’s, inflammatory bowel diseases, and multiple sclerosis [41]. The lucrative non-psychoactive CBD displayed potential therapeutic effects in neuropsychiatric disorders, cancers, and pain management with diverse molecular targets and signalling pathways [42,43,44,45,46,47,48]. Cannabielsoin (CBE) was recently reported to control neuropathic pain via the modulation of the Wnt/β-catenin pathway [49]. Moreover, nine CBC-type and eleven CBN-type PCs were also identified. The potential neuroprotective effect of minor PCs such as CBG, CBDV, CBC, CBN, ∆^9^-THCV, and ∆^9^-THCA was recently reviewed [50]. Thirty miscellaneous PCs, together with nine cannabitriol (CBT)-type PCs, were also isolated and characterised.

Forty-two NC phenols were identified in *Cannabis* and are subclassified into 16 spiro-indans, 12 dihydrostilbenes, 7 dihydrophenanthrenes, and 7 simple phenols [7]. In addition, 34 *Cannabis* flavonoids were also isolated and fully characterised [7]. On the other hand, the second major subclass of NCs is terpenes, where 61 monoterpenes (C10) and 51 sesquiterpenes (C15) were isolated from *Cannabis* with a few reported di- or triterpenoids, as recently reviewed by Radwan et al. [7]. The anticancer effects of *Cannabis* flavonoids and terpenes were reviewed [51] alongside the different biological activities of *Cannabis* phenolics [52]. Various preclinical or in silico studies have outlined the antioxidant, antiparasitic, anti-inflammatory, neuroprotective, and antiviral potential of cannaflavins [53]. 

Alkaloid is another non-cannabinoid subclass that has been detected in *Cannabis* roots. Briefly, two spermidine alkaloids have been isolated and identified—namely, cannabisativine [54] and anhydrocannabisativine [55] in the methanolic root extract or ethanolic leaves/roots extract of *C. Sativa*, respectively. Notably, *Cannabis* roots are not a significant source of cannabinoids, and the currently available data on roots support their ethnobotanical potential in inflammation and pain [56]. 

## 3. Factors Affecting the Extraction Efficiency of Phytochemicals from Plant Materials

Numerous factors have influenced the extraction efficiency of phytochemicals from plant materials, including *Cannabis*. The major influencing factors affecting the extraction efficiency of phytochemicals from plant materials are shown in Figure 3. Understanding these factors can help establish the optimal conditions for the cost-effective extraction of the target compounds from the plant matrix. Plant materials play an important role in the extraction efficiency of their bioactive compounds. Each plant material has specific matrix, structure, and phytochemicals; therefore, the extractability of the phytochemicals varies depending on the type of material. With the same plant material, different parts, such as leaves, stems, roots, and flowers, have different extractability of the target compounds [57]. In addition, fresh, dried, or ground plant materials with small particle sizes have different extraction efficiencies when the extraction is performed under the same conditions [58,59]. Although no specific studies have been reported to compare the impact of different drying conditions and particle sizes on bioactive compounds from *Cannabis*, it is worthwhile to consider future studies for the better selection and preparation of starting materials for the extraction of bioactive compounds.

Solvents are known to directly affect the extraction efficiency of phytochemicals from plant materials. With a range of solvents from polar to nonpolar, various compounds with corresponding polarities can be extracted into solvents, depending on the polarities of the solvents and target compounds. Typically, a combination of polar and less polar solvents is more effective in extracting phytochemicals from plant materials [60]. The volume of solvents or materials and the solvent ratio has been reported to affect the extraction efficiency of phytochemicals from plant materials. Theoretically, the less volume of solvents used, the less extraction efficiency can be obtained because of the saturation [61]. However, a suitable material-to-solvent ratio needs to be applied for cost-effective purposes as more energy is needed to heat the larger volume. In addition, more energy is required to remove water for further concentration or the production of powder.

In some cases, the pH of the solvent has been found to influence the extraction efficiency of the target compounds, as acidic solvents are believed to disrupt the cell walls faster, thus increasing the extraction efficiency. In addition, bioactive compounds are more stable under acidic conditions [61]. Therefore, comprehensive studies are warranted to identify the right solvents, solvent-to-material ratios, and pH for the maximum extraction of phytochemicals from *Cannabis*.

Temperature and length of extraction have been reported to affect the extraction efficiency of phytochemicals from plant materials. A higher temperature with a longer extraction time usually leads to a higher extraction efficiency. However, the stability of phytochemicals can decrease when they are exposed to high temperatures for a long time because most phytochemicals are sensitive to heat [58,61]. Therefore, it is important to determine the most suitable temperature and length of extraction to extract a high level of phytochemicals with minimum degradation. Agitation and pressure have been found to influence the extraction efficiency of phytochemicals from plant materials. Studies have reported that agitation significantly increases the extraction efficiency of phytochemicals compared to non-agitation [62]. Extraction with high pressure has improved the extraction efficiency of phytochemicals [63]. The number of extractions or extraction time has been found to affect the extraction efficiency. The more extraction time is applied for the same quantity of the sample, the more phytochemicals can be extracted [58]. It is similar to the material-to-sample ratio, the use of more solvents for extraction leads to more phytochemicals in plant matrixes that can be released into solvents. However, it should be considered carefully, as more energy is required to heat up the larger solvent volume as well as to remove solvents. Finally, extraction techniques have been reported to significantly affect the extraction efficiency of phytochemicals from plant materials. Extraction techniques can be categorised into two major groups, including conventional extraction techniques and advanced techniques. In some cases, advanced extraction techniques show more efficiency than conventional techniques [64,65]. However, the cost of setting up on a commercial scale is a major limitation for the advanced techniques. Conventional and advanced techniques have been applied for the extraction of phytochemicals from *Cannabis* and are summarised in Table 1 and are further discussed in Section 4.

## 4. Current Techniques for Extraction of Phytochemicals from *Cannabis*

### 4.1. Conventional Extraction of Phytochemicals from Cannabis

*Cannabis* is chemically diverse, with cannabinoids, phenolic compounds, and terpenes being the most important phytochemicals [75]. Several conventional extraction techniques have been applied to extract phytochemicals from *Cannabis* using various solvents [75,76,77,78,79,80,81,82,83]. Different conventional extraction methods, including Soxhlet extraction, maceration, and dynamic maceration, have been employed to extract phytochemicals from *Cannabis* [68,75,76,80,82]. However, studies have also revealed the caveats of using conventional extraction methods, such as the extraction of unwanted substances and degradation of heat-sensitive compounds due to extraction under high temperature [84]. As cannabinoids, phenolic compounds, and terpenes are the three main classes of secondary metabolites in *Cannabis*, the conventional extraction of these main classes is discussed further in the following subsections. The extraction of phytochemicals from *Cannabis* are summarised in Table 1.

#### 4.1.1. Conventional Extraction of Cannabinoids

*Cannabis* contains 125 individual cannabinoids, but Δ^9^-THCA and CBDA are the most predominant [81]. These two compounds undergo decarboxylation to produce Δ^9^-THC and CBD [81,85] *(*Figure 1*)*. Ethanol has been found as an effective solvent in extracting cannabinoids using hot maceration [68,75] as well as Soxhlet extraction [86]. However, the extraction efficiency of these two conventional techniques was significantly lower than that of the advanced microwave-assisted extraction [86]. Although conventional extraction techniques have advantages such as simple procedures, easy operations, and affordability, these conventional techniques have several drawbacks, including longer extraction times and the demand for larger solvent volumes, leading to an inclined overall operation cost and harmful environmental impact compared to modern techniques [80]. Furthermore, a stable high temperature of Soxhlet extraction has also been reported to accelerate the degradation of THCA to THC and, finally, to CBN, leading to high levels of THC and CBN in the extract [70,80]. A recent study extracted CBD from inflorescences of *C. sativa* using the methanol solvent maceration technique and supercritical fluid extraction technique and found that conventional extraction obtained a higher oil yield but lower CBD in comparison with supercritical fluid extraction [81]. 

Another conventional extraction technique called dynamic maceration has also been applied to extract cannabinoids, mostly from industrial hemp [79,80]. Dynamic maceration is a solid-lipid extraction procedure where a sample is soaked in organic solvents that were selected based on the polarity of the target compounds at a specific temperature for a specific time and then agitated [79,80]. Ethanol, acetonitrile, and hexane are the common solvents used to extract cannabinoids [87,88,89]. Ethanol was reported to be more efficient in extracting acidic cannabinoids using dynamic maceration than other organic solvents (hexane, acetone, and methanol); however, for neutral cannabinoids, ethanol is on par with other organic solvents [79]. Ethanol was also found to be more effective than other organic solvents in different studies [73,90]. Furthermore, organic solvent mixtures such as methanol and chloroform at 9:1 (*v/v*) were effective to recover cannabinoids from *C. sativa* under constant agitation [91]. Interestingly, Romano and Hazekamp [92] discovered that olive oil is more effective in extracting *Cannabis* oils containing cannabinoids and terpenes than ethanol, which also extracts chlorophylls, imparting a distinct green colour and unpleasant taste in the final product. The cannabinoid degradation rate was found to be much slower in olive oil extracts as compared to ethanolic extracts [80,93]. 

#### 4.1.2. Conventional Extraction of Phenolic Compounds

Three major classes of phenolic compounds, flavonoids, stilbenoids, and lignans, have been identified in *Cannabis* [52,75,94]. Acetone, methanol, ethanol, and their aqueous mixtures are the common organic solvents to extract phenolic compounds from industrial hemp. Most of these studies quantified the phenolic compounds of the hemp extracts by the overall determination of the total phenolic content and total flavonoid content. In addition, several studies have quantified individual phenolic compounds, such as caffeic acid, gallic acid, rosmarinic acid, *p*-OH-benzoic acid, ferulic acid, 3,4-dihydroxybenzoic acid, *p*-coumaric acid, syringic acid, quercetin, luteolin, canniprene, cannflavin A, cannflavin B, catechin, naringenin, isorhamnetin, resveratrol, rutin trihydrate, apigenin, and apigenin 7-glucoside in the extracts [95,96,97,98]. Of note, most conventional extraction was conducted at room temperature and not under optimal conditions [99,100,101,102,103]. Furthermore, an aqueous solution of 2-hydroxypropyl-β-cyclodextrin (a green solvent) has also been applied to recover phenolic compounds from the by-products of industrial hemp oil processing [103]. 

#### 4.1.3. Conventional Extraction of Terpenes

Terpenes such as α-pinene, β-pinene, β-myrcene, limonene, terpinolene, linalool, α-terpineol, β-caryophyllene, α-humulene, and caryophyllene oxide are known as the major constituents of *Cannabis* essential oils [72,75]. Similar to cannabinoids and phenolic compounds, most studies reported in the literature have been performed on the industrial hemp variety of *C. sativa* [104,105,106]. Distillation techniques such as hydro-distillation and steam distillation have been implemented to extract essential oils (terpenes) from *Cannabis.* However, they were less effective than the advanced supercritical fluid extraction at a lower temperature [105]. Interestingly, the GS–MS analyses in the same study revealed that the steam distillation at 130 °C and hydro-distillation at 110 °C showed 37 and 35 terpenes, respectively, in the essential oil of *C. Sativa*, whereas essential oil extracted with supercritical fluid at 45 °C showed only 30 terpenes [105], indicating the generation of terpenoid artefacts. 

Solvent-based (both polar and nonpolar) conventional extraction methods have also been implemented to isolate terpenes from *Cannabis* [66,67,75,107,108]. For example, Ibrahim et al. [72] used an ethyl acetate, ethanol, methanol, and chloroform/methanol (1:9; *v/v*) mixture and reported that ethyl acetate was the best solvent to recover terpenes from *Cannabis* [72]. However, mixtures of organic solvents are generally more effective for extracting terpenes from *Cannabis* than individual solvents. For instance, a study on the inflorescences of *C. sativa* found that the hexane and ethanol mixture at 7:3 (*v/v*) is more efficient in extracting terpenes compared to hexane or ethanol alone [67]. 

Fischedick et al. [66] used ethanol as the extraction solvent to recover both terpenes and cannabinoids, and 36 compounds were identified in 11 varieties of *C. sativa* [66]. Conventional ethanolic extraction was also used by A. Hazekamp and Fischedick [88] to isolate monoterpenes (α–pinene, myrcene, and terpinolene); sesquiterpenes ((*E*)–caryophyllene and α–humulene) [88]; and oxygenated terpenes (guaiol, γ_eudesmol, and α_bisabolol) from marijuana and medical *Cannabis* inflorescences [82]. 

Six different preparation methods were employed in a recent study to obtain *Cannabis* oils with high cannabinoids and terpene contents [109]. In this study, conventional extraction with olive oil or ethanol, along with steam distillation, was included. In conclusion, it was recommended that *Cannabis* should be macerated at room temperature to obtain the optimal terpene and cannabinoid yields in *Cannabis* oil [109]. Interestingly, a previous study reported that drying methods also influence the terpene composition of *Cannabis* extract, and gentle drying with a nitrogen stream can retain monoterpenes and sesquiterpenes in the *Cannabis* extract [82]. Thus, the dehydration of samples before extraction should be optimised to prevent the degradation of terpenes. 

In summary, although various conventional extraction techniques have been applied to extract terpenes, cannabinoids, and phenolic compounds from *Cannabis*, most studies have primarily focused on the industrial hemp variety without optimisation. Therefore, further investigations are needed to optimise the conventional extraction parameters, such as type of solvent, sample-to-solvent ratio, temperature, and time, which could benefit the recovery of these compounds from both industrial hemp and medicinal marijuana. The implementation of mathematical prediction models is also recommended to facilitate the optimisation process to obtain greater yields of bioactive compounds in *Cannabis* extracts. Impacts of the variations related to season; geographical location; and *Cannabis* type (strain, chemotypes, and chemovars) on the recovery of these compounds in *Cannabis* extracts or oils should also be considered while designing future studies.

### 4.2. Advanced Extraction Techniques

Open-loop and closed-loop systems with butane hash oil (BHO) have been applied to extract terpenes with more flavours and aroma from *Cannabis*. However, the closed-loop system is often used, as it is much safer and with more advantages than the open-loop setting [110]. BHO is one of the cheapest and efficient solvents that offer the most desired final product. However, its most common disadvantage is being hard to handle in bigger batches, as it is highly flammable, colourless, and odourless [111]. Therefore, the legal use of BHO is restricted to licensed producers [112]. The closed-loop system consists of a butane reservoir, trim tube, evaporation chamber, vacuum pump, recovery pump, vacuum oven, and gas detectors. This system provides a more stable and environmentally friendly platform for dealing with volatile BHO. 

A multi-solvent extraction system, such as PX 40, has been applied to effectively extract phytochemicals from *Cannabis*. PX 40 typically functions using either butane or propane or a mixture of both. Despite its high cost, it is a productive system [113]. In addition, the pressurised liquid extraction (PLE) technique has been applied to extract phytochemicals from *Cannabis*. Common solvents like water, methanol, acetone, and hexane have been applied to extract the phytochemicals for a short time under high temperature and pressure (temperature range of 75–150 °C and pressure of usually 10.4 MPa). In this technique, when increasing the temperature, the internal pressure in the cell is consequently increased and push the components to the outside of the cell through the cell wall pores [114]. Subcritical carbon dioxide (CO_2_) extraction systems with specialised pressure and controlled temperature have also been applied to extract high-quality cannabinoids, terpenes, and flavonoids. This technique is an effective nonpolar extraction method [115]. In addition, carbon (IV) oxide is also an effective method to extract CBD oil, with up to a 90% extraction efficiency [116].

Supercritical fluid extraction with CO_2_ (SFE-CO_2_) has been applied to extract phytochemicals from *Cannabis*. This technique is more effective compared to subcritical CO_2_ extraction systems [115]. It should be noted that solvents used and other factors, such as temperature, pressure, and sample types, can affect the extraction yield of this technique. The best conditions for extracting phytochemicals were found with a back pressure of 12 MPa, a flow rate of 10 mL/min CO_2_, and a pump rate of 1 mL/min, with a temperature of 25 °C. The solvent gradient conditions were 100−50% solvent A and 0−50% solvent B, and the time was 0–25 min, and all the extractions were run in two cycles [117,118,119]. When scaling up in a pilot-scale using SFE-CO_2_ (SCF100 model 3 PLC-GR-DLMP, Separeco S.r.l, Pinerolo, Italy), 10 MPa pressure and a temperature of 40 °C were the best conditions [120]. 

Other advanced extraction techniques, such as dynamic maceration (DM), ultrasound-assisted extraction (UAE), and microwave-assisted extraction (MAE), have been applied to extract phytochemicals from *Cannabis* [73]. The DM technique focuses on maceration of the *Cannabis* in organic solvent, then concentrating the extracted solution by removing the solvent under reduced pressure, high temperature, and acid. However, the chemical structures of the final target compounds can change during the extraction process [68].

MAE and UAE have been found to effectively extract phytochemicals from *Cannabis* [73,121]. Different factors can influence the extraction efficiency of phytochemicals from *Cannabis* using UAE and MAE, such as the type of solvent, sample size, sample-to-solvent ratio, time, and power. However, none of the previous studies comprehensively compared and established the optimal conditions for these two techniques when extracting cannabinoids, phenolics, or terpenes from *Cannabis*. 

The nonthermal pulsed electric field-assisted technique (PEF) has been applied to extract oil from *Cannabis* seeds. This method is effective, with less risk of thermal degradation of the phytochemicals [122]. PEF has been found to have higher extraction efficiency with better oil quality [74,123,124]. However, the main disadvantage is that PEF requires high-voltage pulses [125]. 

The enzyme-assisted extraction technique (EAE) has also been utilised to extract oil from *Cannabis*. Enzymes like pectinase or cellulose, either alone or in mixtures, are applied during the extraction process to release phytochemicals from the plant cells [126] Kitrytė et al. [127] reported that EAE increased the recovery of cannabinoids from *Cannabis* by 20.2% (*w/w*) of the water-soluble constituents and 94% of mono- and disaccharides [127]. There are two EAE techniques, including enzyme-assisted cold pressing and enzyme-assisted aqueous extraction [128]. Although EAE is an expensive technique, it is very promising for the effective extraction of phytochemicals from *Cannabis*. The extraction of phytochemicals from *Cannabis* are summarised in Table 1.

## 5. Future Considerations for Effective Extraction of Bioactive Compounds from *Cannabis*

As *Cannabis* is a rich and diversified source of phytochemicals, it would be challenging to have an optimal extraction technique or condition to recover all bioactive molecules from it. Although numerous conventional and advanced techniques have been employed to extract phytochemicals from *Cannabis*, as discussed in Section 4, there are still gaps for future studies to understand more about the materials, the impact of different extraction parameters, and the stability of phytochemicals derived from *Cannabis*. Therefore, future studies should focus on the gaps summarised in Figure 4. As discussed in Section 4, no comprehensive studies have compared the phytochemicals, including cannabinoids, phenolics, and terpenes, between different species and their different parts, such as leaves stems, flowers, and roots. In addition, the influence of geographical locations, harvesting season, harvesting time, and cultivation methods on the *Cannabis* phytochemicals needs to be considered to have high-quality starting materials for extraction. Systematic investigations are also necessary to understand the impact of different transportation, storage, and drying conditions and methods on the retention of cannabinoids, phenolics, and terpenes from *Cannabis*. As these phytochemicals are sensitive, storage and transportation under appropriate conditions are crucial. In addition, drying can affect phytochemicals from *Cannabis* [82], and further studies on different drying conditions and methods are needed to identify the best drying conditions and methods for *Cannabis* before the extraction process.

Various conventional and advanced techniques have been applied to extract phytochemicals from *Cannabis*. However, most of these studies have focused on the individual impact of a few extraction parameters and have not considered all the possible factors and their interactive effects. Future studies are recommended to consider all possible parameters, as discussed in Section 3, and determine their individual and interactive effects on the extraction efficiencies of different phytochemicals from *Cannabis*. A comparison of the effectiveness of different extraction techniques under their optimal conditions is also needed to generate clear scientific evidence on the most effective extraction techniques and conditions.

It is typically complicated to study multiple factors and their interactive effects. Therefore, mathematical models such as response surface methodology are recommended to design, predict, and obtain the desired outcomes. Future studies should consider the cost-effectiveness of extraction techniques as well for commercial applications. We highlighted that extraction efficiency is not the sole criteria for the selection of the appropriate extraction technique, where the cost for extraction is equally important. Finally, more studies are needed to isolate compounds in groups or individually from *Cannabis*, test their properties, and propose further applications as therapeutic agents or functional ingredients.

## 6. Conclusions

Despite the growing interest in *Cannabis* and its bioactivity, as demonstrated in several preclinical and clinical studies, reports optimising the extraction techniques and factors to recover bioactive compounds such as terpenes, cannabinoids, and phenolic compounds from the medicinal marijuana variety of *C. sativa* are limited in the literature, with most studies utilising the industrial hemp variety. This limitation might be attributed to the limited availability of medicinal marijuana due to the regulations imposed in different countries on its cultivation and utilisation. However, previous studies have demonstrated that the yield and bioactivity of phytochemicals are primarily influenced by the extraction techniques implemented. As aforementioned, several factors such as the types of plant materials, extraction techniques/time, solvents, pH, temperature, pressure, and material-to-solvent ratio can influence the extraction efficiency of bioactive compounds from plant matrices. Therefore, further investigations to optimise these factors could be advantageous to recover bioactive compounds from both industrial hemp and medicinal marijuana. As the optimisation of these factors can be highly expensive, time-consuming, and labour-intensive, the utilisation of mathematical prediction models could accelerate the optimisation process for both conventional (e.g., Soxhlet extraction, maceration, hydro-distillation, and steam distillation) and advanced extraction techniques. The most common extracted methods used in the *Cannabis* industry are pressurised liquid extraction, subcritical CO_2_ extraction, supercritical fluid CO_2_ extraction, nonthermal pulsed electric field-assisted, MAE, UAE, and enzyme-assisted extraction. Furthermore, the impacts of the variations related to season; geographical location; and *Cannabis* type (strain, chemotypes, and chemovars) on the recovery of phytochemicals in *Cannabis* extracts or oils should also be considered whilst designing future studies.

## Figures and Tables

**Figure 1 molecules-27-00604-f001:**
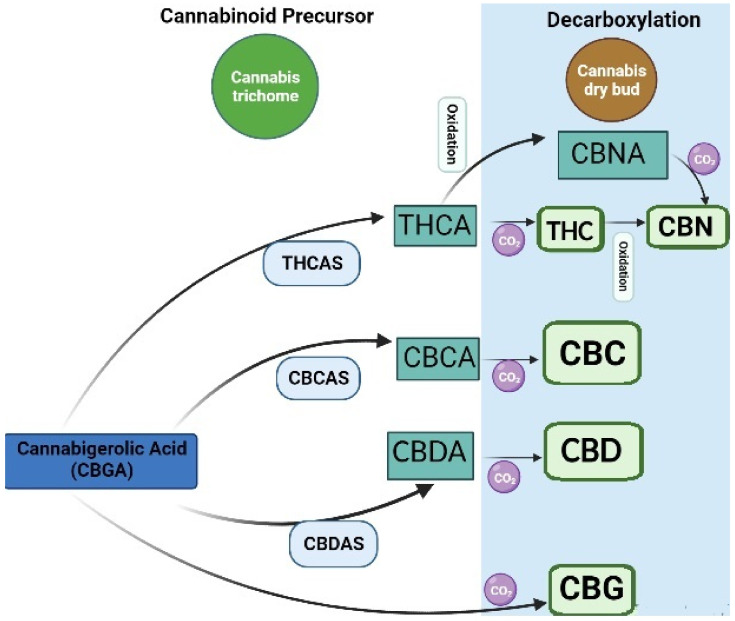
Biosynthesis of cannabinoids: cannabigerolic acid; CBGA, tetrahydrocannabinolic acid synthase; THCAS, cannabidiolic acid synthase; CBDAS, cannabichromene acid synthase; CBCAS, tetrahydrocannabinolic acid, THCA; cannabidiolic acid; CBDA, cannabichromenic acid; CBCA, cannabinolic acid; CBNA, cannabichromene; CBC, cannabidiol; CBD, cannabigerol; CBG, cannabinol; CBN, tetrahydrocannabinol; THC.

**Figure 2 molecules-27-00604-f002:**
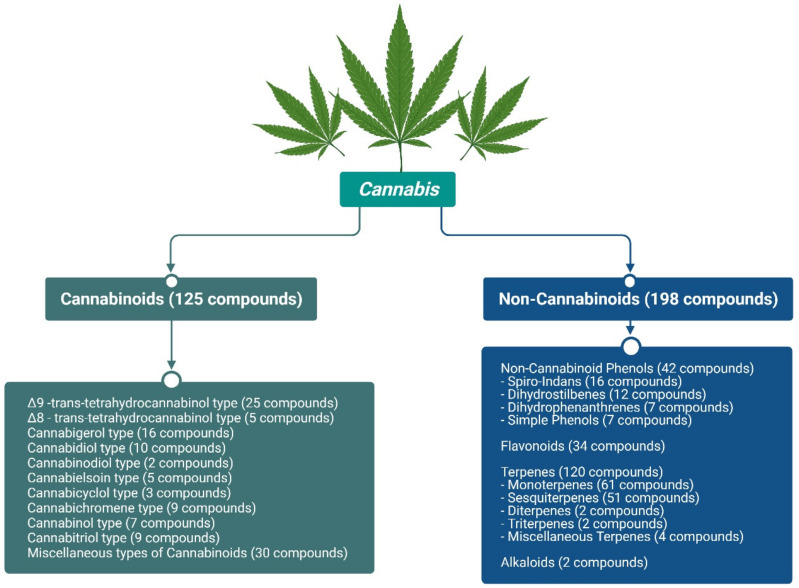
Major phytochemicals identified from cannabis.

**Figure 3 molecules-27-00604-f003:**
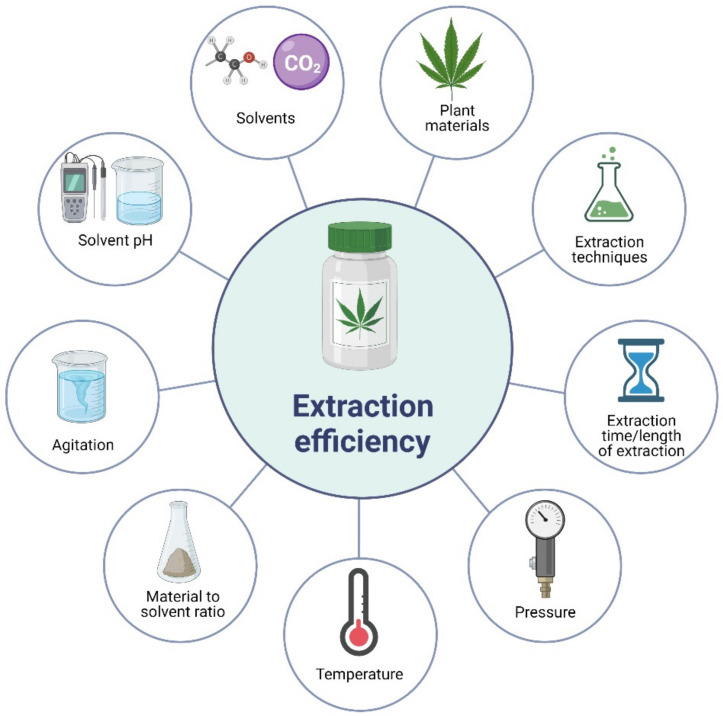
Factors affecting the extraction efficiency of plant bioactive compounds.

**Figure 4 molecules-27-00604-f004:**
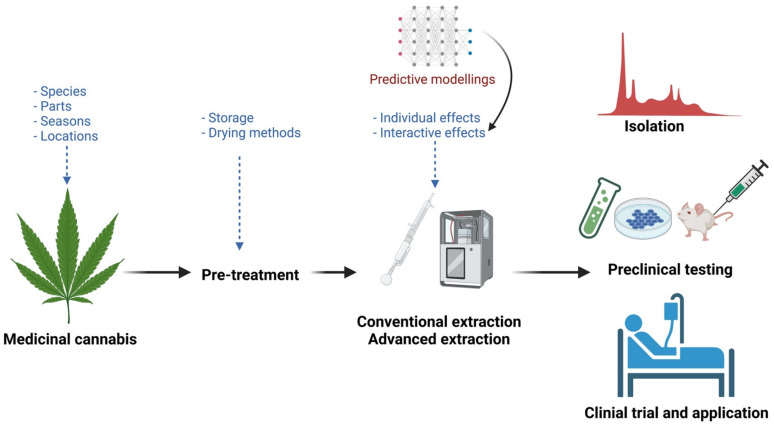
Proposed gaps for future studies.

**Table 1 molecules-27-00604-t001:** Extraction of phytochemicals from *Cannabis*.

Extraction Technique	Extraction Conditions/Procedures	Advantages and Limitations	References
Solvent extraction	The plant materials (0.9–1.1 g) were crushed and extracted in 45-mL ethanol for 15 min with the agitation of 400 rpm. Extracts were centrifuged briefly for 30 s at 2000 rpm. The supernatant was collected and filtered.	A simple technique, but not very efficient	[66]
Solvent extraction	Samples were extracted in hexane and ethanol mixture at 7:3 (*v*/*v*) and shaken for 45 min at 225 rpm in a TU-400 orbital shaker incubator at room temperature to obtain the extract.	A simple technique, but not very effective	[67]
Solvent extraction	Samples were extracted in ethanol at room temperature for 45 min to obtain the extract.	A simple technique, but not very effective	[68]
Solvent extraction	The plant material (100 g) was Pulverised and extracted with 500-mL petrol ether acidified with acetic acid (0.5-mL CH_3_COOH in 500-mL PE). The filtrated extract was re-extracted 3 times with 400 mL of NaOH and Na_2_SO_3_ (2% each). These combined extracts were acidified with 500 mL of 5% sulfuric acid until pH reached 3 and immediately extracted 3 times with 400-mL TBME. These combined organic extracts were dried with Na_2_SO_4_, filtrated, and concentrated in a rotary evaporator at 25–30 °C with cryostatic cooling of the vapours. The concentrate was dried overnight at vacuum conditions, yielding 1.71-g brown amorphous material.	A simple technique, but not effective and difficult for commercial production	[69]
Soxhlet extraction	Ground dried samples (2 g) were extracted using Soxhlet extractor for 1, 2, or 3 h with 75 mL of n-hexane or methanol then cooled to room temperature to obtain the extract.	A simple technique, but not effective	[70]
Sonication	The dried and pulverised plant material (50 g) was extracted by sonication and periodic shaking (30 min) with 250-mL petroleum ether, which was acidified with 0.5-mL concentrated acetic acid. The extract was further extracted 3 times with 200 mL of an aqueous solution (2% *w/v* each) of sodium hydroxide and sodium sulphite. The combined and cooled water phases were acidified with about 250-mL cooled sulphuric acid to pH 3 and immediately extracted 5 times with 200-mL diethyl ether. The combined organic phases were dried with sodium sulphate and evaporated to dryness.	Quite effective advanced technique, but it is challenging to apply on a commercial scale	[71]
Sonication	Samples (1 g) were extracted with 10 mL of the extraction solution (100 μg/mL of n-Tridecane in ethyl acetate) by sonication for 15 min to obtain the extract.	An advanced technique, but not under optimal conditions	[72]
Ultrasound-assisted extraction (UAE)	A small amount (0.25 g) of the sample was mixed with 10 mL of ethanol and was then extracted 3 times using UAE at 40 °C for 15 min. The solution was then filtered through a paper filter to obtain the final extract.	An advanced technique, but not under optimal conditions	[73]
Pressurized liquid extraction (PLE)	Samples of *Cannabis* (0.3 g) were mixed with sand and then placed into a 22-mL stainless-steel extraction cell with a cellulose filter. The sample cells were then closed and placed into the carousel of the ASE 200 system. Methanol or n-hexane was used as extraction solvents. Extractions were carried out at 25, 50, 75, 100, 125, and 150 °C at a pressure of 40 bar. Extractions were performed for 5, 10, 15, or 20 min. After the extraction process, the extraction cell content was flushed with the same solvent in the amount equal to 60% of the extraction cell volume and purged for 60 s by applying pressurized nitrogen (at 150 psi) to obtain the final extracts.	An advanced technique, but it has not been operated under optimal conditions.	[70]
Ultrasound-assisted extraction (UAE)	A small amount (0.25 g) of sample was mixed with 10 mL of ethanol and was then extracted 3 times using UAE at 40 °C for 15 min. The solution was then filtered through a paper filter to obtain the final extract.	An advanced technique, but not under optimal conditions	[73]
Pulse electric field extraction (PEF)	The seeds were treated by the PEF process (0, 3, and 6 kV/cm). The PEF process was conducted with a capacity of a process chamber of 4 L. Maximum voltage of the instrument was 7 Kv, and its capacitance was 8 μF.	An advanced technique, but not under optimal conditions	[74]

## Data Availability

Not applicable.

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
