# Peer review of "A Comprehensive Review on the Techniques for Extraction of Bioactive Compounds from Medicinal Cannabis"

_molecules, 2022, doi:10.3390/molecules27030604_

Round 1
Reviewer 1 Report
This is an interesting, well documented, well illustrated and clearly written review dealing with the extraction methods of the numerous compounds present in cannabis.
My only suggestion is to recall all the abbreviations in Fig. 1 in the legend below. It would be too difficult to search for each in the text.
Author Response
My only suggestion is to recall all the abbreviations in Fig. 1 in the legend below. It would be too difficult to search for each in the text.
Response: Thanks for the positive comments. All the abbreviations were spelled out in the legend of Figure 1 which could be tracked in the revised version of the manuscript.
Reviewer 2 Report
The authors Hebah Muhsien Sabiah AL Ubeed et al in the article titled " A comprehensive review on the techniques for extraction of bioactive compounds from medicinal cannabis" have made an extensive review on the extraction methods, critical factors affecting extraction yields, and major bioactive compounds in cannabis, as well as the future consideration for the effective extraction of bioactive compounds from cannabis.
It is an excellent read and I commend the authors on such an extensive literature review.
Some suggestions for authors to further improve the MS are as follows.
- Figure 1 shows the biosynthesis of cannabinoids, ‘Extraction’ should be removed.
- The detailed chemical information of different bioactive compounds could be summarized in a table.
Author Response
The authors Hebah Muhsien Sabiah AL Ubeed et al in the article titled " A comprehensive review on the techniques for extraction of bioactive compounds from medicinal cannabis" have made an extensive review on the extraction methods, critical factors affecting extraction yields, and major bioactive compounds in cannabis, as well as the future consideration for the effective extraction of bioactive compounds from cannabis.
It is an excellent read and I commend the authors on such an extensive literature review.
Response: Thanks for the positive comments.
Some suggestions for authors to further improve the MS are as follows.
Figure 1 shows the biosynthesis of cannabinoids, ‘Extraction’ should be removed.
Response: Amended as directed, thanks.
The detailed chemical information of different bioactive compounds could be summarized in a table.
Response: Thanks for the suggestions. We do not think a Table is necessary because the detailed chemical information of different bioactive compounds was already summarised in Figure 2 (Page 4). In addition, chemical information was also well-reviewed by a recent review in Molecules by Radwan et al. 2021 (Radwan, M.M.; Chandra, S.; Gul, S.; ElSohly, M.A. Cannabinoids, phenolics, terpenes and alkaloids of cannabis. Molecules 2021, 26, 2774.) which we also cited in our current manuscript.
Reviewer 3 Report
Page 3 and 4 Page Figures need to be properly named there are missing figure designation. Page 4 sentience „Figure 1. Major phytochemicals identified from cannabis „ are this title of the section or missing designation of the Figure above.
In section 2 of manualscript “ 2. Major cannabis phytochemicals and their therapeutic effects” there are mentions Figures listed below but there are not presented in manuscript.
(Figure 2), (Figure S1-S3), Figure S2, Figure S4, (Figure S5), (Figure S6).
there are also several chemical compounds that are listed numerically but not for example (Compounds 67-75) what are these compounds, and so on (Compounds 76-86), (Compounds 96-125), (Compounds 87-95), (Compounds 126-141), (Compounds 142-153), (Compounds 154-160).
Section 4
The review paper should provide an overview of the extraction techniques and conditions used. Summarized so that the reader can see which parameters were used and what the effects of these parameters were in the extraction process.
Author Response
Page 3 and 4 Page Figures need to be properly named there are missing figure designation. Page 4 sentience „Figure 1. Major phytochemicals identified from cannabis „ are this title of the section or missing designation of the Figure above.
Response: Both the Figures have been properly titled and it's been designated in the revised manuscript.
In section 2 of manualscript “ 2. Major cannabis phytochemicals and their therapeutic effects” there are mentions Figures listed below but there are not presented in manuscript.
There are also several chemical compounds that are listed numerically but not for example (Compounds 67-75) what are these compounds, and so on (Compounds 76-86), (Compounds 96-125), (Compounds 87-95), (Compounds 126-141), (Compounds 142-153), (Compounds 154-160).(Figure 2), (Figure S1-S3), Figure S2, Figure S4, (Figure S5), (Figure S6).
Response: We have deleted theseTYPOs. Thank you.
Section 4
The review paper should provide an overview of the extraction techniques and conditions used. Summarized so that the reader can see which parameters were used and what the effects of these parameters were in the extraction process.
Response: We would like to thank the reviewer for the constructive suggestion. We have added a new Table (Table 1), which summarized the extraction process in the revised manuscript to provide comprehensive information to the readers.
Round 2
Reviewer 3 Report
Dear authors, the natural changes have significantly contributed to the clarity and intelligibility of the work. The work is well thought out, and well structured. It is necessary to refine the table that has been added in order to be reviewed and easy to read. The table in its current form is not clear and difficult to follow.
Author Response
We have significantly revised the Table to address the previous extraction techniques, their procedures as well as their advantages and limitations. We believe that the revised Table now is informative and easy to follow by the readers. Thank you very much.
